# Alcohol Contribution to Total Energy Intake and Its Association with Nutritional Status and Diet Quality in Eight Latina American Countries

**DOI:** 10.3390/ijerph182413130

**Published:** 2021-12-13

**Authors:** Juan Carlos Brenes, Georgina Gómez, Dayana Quesada, Irina Kovalskys, Attilio Rigotti, Lilia Yadira Cortés, Martha Cecilia Yépez García, Reyna Liria-Domínguez, Marianella Herrera-Cuenca, Viviana Guajardo, Regina Mara Fisberg, Ana Carolina B. Leme, Gerson Ferrari, Mauro Fisberg

**Affiliations:** 1Neuroscience Research Center, Institute of Psychological Research, University of Costa Rica, San José 10501-2060, Costa Rica; 2Departament of Biochemistry, School of Medicine, University of Costa Rica, San José 11501-2060, Costa Rica; georgina.gomez@ucr.ac.cr (G.G.); dahiana.quesada37@gmail.com (D.Q.); 3Career of Nutritión, Faculty of Medical Sciences, Pontificia Universidad Católica Argentina, Buenos Aires C1107AAZ, Argentina; ikovalskys@gmail.com; 4Departament of Nutrition Diabetes and Metabolism, School of Medicine, Faculty of Medicine, Pontificia Universidad Católica de Chile, Santiago 8330024, Chile; arigotti@med.puc.cl; 5Departament of Nutritión and Biochemistry, Pontificia Universidad Javeriana, Bogotá 110231, Colombia; ycortes@javeriana.edu.co; 6School of Health Sciences, San Francisco University of Quito, Quito 17-1200-841, Ecuador; myepez@usfq.edu.ec; 7Institute of Nutritional Research, Lima 15026, Peru; rpareja@iin.sld.pe; 8Career of Nutrition and Dietetics, Faculty of Health Sciencies, Peruvian University of Applied Sciences, Lima 15067, Peru; 9Center for Development Studies, Bengoa Fundation of Food and Nutrition, Central University of Venezuela, Caracas 1053, Venezuela; manyma@gmail.com; 10Institue for Scientific Cooperation in Health and Environment, Buenos Aires Santa Fe Av. 1145, Caba C1059ABF, Argentina; viviana.guajardo@comunidad.ub.edu.ar; 11Department of Nutrition, School of Public Health, University of São Paulo, Sao Paulo 01246-904, Brazil; regina.fisberg@gmail.com (R.M.F.); acarol.leme@gmail.com (A.C.B.L.); 12Center for Nutrology and Feeding Difficulties, Pensi Institute, José Luiz Egydio Setubal Foundation, Sarabá Children’s Hospital, Sao Paulo 01228-200, Brazil; mauro.fisberg@gmail.com; 13Family Relations and Applied Nutrition, University of Guelph, Guelph, ON N1G 2W1, Canada; 14Sciences of Physical Activity, Sports and Health School, Universith of Santiago of Chile, Santiago 7500618, Chile; gerson.demoraes@usach.cl; 15Departament of Pediatrics, School of Medicine, Federal University of Sao Paulo, São Paulo 04023-061, Brazil

**Keywords:** alcohol intake, macronutrients, micronutrients, food groups, Latin America, nutrition survey

## Abstract

Alcohol consumption is a modifiable risk factor for non-communicable diseases. This study aimed to characterize alcohol consumers at the nutritional, anthropometric, and sociodemographic levels. Data from 9218 participants from Argentina, Brazil, Chile, Colombia, Costa Rica, Ecuador, Peru, and Venezuela participating in “Latin American Health and Nutrition Study (ELANS)”, a multi-country, population-based study, were used. Dietary intake was collected through two, 24 h recalls. Participants were classified into consumers (*n* = 1073) and non-alcohol consumers (*n* = 8145) using a cut-off criterium of ≥15 g/day of alcohol consumption calculated from the estimation of their usual daily intake. Among alcohol consumers, the mean alcohol consumption was 69.22 ± 2.18 grams (4.6. beverages/day), contributing to 484.62 kcal, which corresponded to 16.86% of the total energy intake. We found that the risk of alcohol consumption was higher in young and middle-aged men from low and middle socioeconomic status. Argentine, Brazil, and Chile had the highest percentage of consumers, while Ecuador showed the highest alcohol consumption. Alcohol drinkers were characterized by having higher body weight and wider neck, waist, hips circumferences. Alcohol drinkers had a higher energy intake, with macronutrients providing relatively less energy at the expense of the energy derived from alcohol. Alcohol drinkers showed lower and higher consumptions of healthy and unhealthy food groups, respectively. In addition, adequacy ratios for all micronutrients assessed were lower in alcohol consumers. All these deleterious effects of alcohol on nutritional and anthropometric parameters increased with the number of alcoholic beverages consumed daily. Altogether, these findings suggest that limiting alcohol consumption can contribute to reducing the risk of obesity, metabolic syndrome, and diet-related diseases.

## 1. Introduction

Historically, alcoholic beverages have been part of the human diet for cultural, social, and spiritual reasons [1]. According to World Health Organization, in 2016, approximately 43% of the worldwide population who were 15 years old or above had consumed alcohol in the previous 12 months, with an average consumption of pure alcohol of 6.4 L per capita [2]. Spirit beverages are the most consumed type of alcoholic beverage (44%), while beer and wine represent the second and third most consumed alcoholic drinks representing 34.3% and 11.7% of total alcohol consumption, respectively. The Americas and European regions exhibited the highest alcohol consumption in the world [2,3]. Population studies reported higher frequencies of alcohol consumption in men than women, while factors such as age, socioeconomic status (SES), and educational level vary among regions [4,5,6]. The excessive consumption of alcohol has been linked to adverse health consequences with high economic costs both at the individual and societal level [3]. Alcohol consumption is considered a risk factor for early death, causing approximately 3 million deaths worldwide each year [7]. It is also a modifiable risk factor associated with a wide range of non-communicable diseases, such as cardiovascular and liver diseases, cancer, neurological and psychiatric disorders (e.g., neurocognitive and drug dependence disorders), and unintentional injuries [7,8]. However, one of the most important yet neglected factors is the impact that alcohol may have on body weight gain, obesity, metabolic syndrome, and diet-related diseases [9]. Alcohol provides 7 kcal/g, making it the second most important source of energy in the diet, surpassed only by fats, which have about 9 kcal/g [10]. It is expected, therefore, that the energy derived from alcohol consumption will sum up to other energy sources promoting a positive energy balance potentially leading to weight gain [11,12]. Furthermore, alcohol consumed before or with meals induces orexigenic effects by increasing appetite and reducing satiation due to changes in the rewarding perception of food [1]. The alcohol-driven, inhibitory control impairment may be another mechanism explaining higher food intake when drinking alcohol before or during meals [12]. 

Despite all this evidence, the relationship between alcohol consumption and obesity is still unclear. While some studies have reported that mild to moderate alcohol intake is associated with a lower prevalence of obesity, others have found quite the opposite. Evidence suggests that alcohol type, consumption patterns, and sex affect the relationship between alcohol and obesity [13]. Sayon et al. (2011) reported that at least seven beers and spirits per week, but not wine, increased the risk for overweight and obesity [10]. In the United States population, it was found that men, but not women, consuming 5 drinks per day increased body mass index (BMI) and waist circumference compared to those consuming 1 to 2 drinks per day [6]. In contrast, alcohol consumption has been associated with a reduced risk of clinical events showing positive effects on cardiovascular health and lower mortality, perhaps by improving the lipid profile and insulin sensitivity among moderate drinkers [14,15]. These positive outcomes of alcohol consumption seem to be dependent on the amount and frequency of alcohol consumption. In Japanese male drinkers, the prevalence of metabolic syndrome and its risk factors increase with alcohol consumption. However, the metabolic syndrome prevalence was significantly higher among the nondrinkers when compared to light or moderate drinkers [14,15]. In Korean men, a consumption greater than 2–3 times per week increased the likelihood of suffering metabolic syndrome, but no associations were found in those drinking less than 2–3 times per week [16]. One other study showed that consuming two or more alcoholic beverages per drinking session was enough to increase waist circumference, triglycerides levels, blood pressure, and fasting plasma glucose [16]. The type of alcoholic beverage appears to have a role in nutritional and health outcomes. For instance, the consumption of beer and spirits, but not wine, increased the risk for obesity and produced greater weight gain after a median follow-up of six years [17]. However, many of these studies did not analyze in detail the dietary habits of their participants, something that should be evaluate to elucidate the contribution of alcohol consumption to the nutritional and health status [18]. This may have compromised identifying a clear cause-and-effect association between alcohol consumption and overweight parameters. Considering that both obesity and excessive alcohol consumption are of public health concern, a better understanding of the association between alcohol consumption and bodyweight gain is warranted.

Latin America and the Caribbean regions have the fifth-highest adult alcohol per capita (APC) (6.8 L) in the world, with most (61%) countries in the Americas exhibiting a total adult APC that is higher than the global average [7]. Coincidently, a representative study with urban samples of Latin America countries showed that ~60% of the population was overweight or obese [19]. To our knowledge, however, there are no studies that have evaluated the relationship between alcohol consumption and the risk of obesity in the Latin American region. The purpose of the present study was, therefore, to determine the drinking patterns of current alcohol consumers according to sociodemographic variables (i.e., country, sex, age, and SES) and the association between alcohol consumption and anthropomorphic measurements (i.e., BMI, body weight, neck, waist, and hips circumference), food intake (e.g., macro- and micronutrients intake and contribution to total energy intake and preferred food groups consumed) and diet quality (e.g., diet quality score and diversity indexes and micronutrients adequacy) in a large, multinational representative sample of urban areas of eight Latin American countries (Argentina, Brazil, Chile, Colombia, Costa Rica, Peru, and Venezuela). 

## 2. Materials and Methods

### 2.1. Study Population

The Latin American Study of Nutrition and Health (Estudio Latino Americano de Nutrición y Salud, ELANS) aimed to assess weight status, food consumption, and physical activity in a representative sample of urban areas of eight Latin American countries (Argentina, Brazil, Chile, Colombia, Costa Rica, Peru, and Venezuela). ELANS is a cross-sectional, household-based multinational survey. A total of 9218 participants aged 15–65 years old were included by a random complex selection of primary and secondary sampling units. The households were selected within each secondary sampling unit and through systematic randomization. Selection of respondents within a household was conducted, including 50% of the sample next birthday and 50% last birthday methods [20], controlling quotas for sex, age, and SES. SES was classified into low, middle, and high, using a country-dependent questionnaire based on governmental reference according to national population census and on legislative requirements or established local standard layouts. Data collection was conducted from September 2014 to July 2015. Details have been previously published [21]. The ELANS protocol was approved by the United States Western Institutional Review Board (#20140605) and the Ethics Review Boards of each of the participating institutions and was registered at Clinical Trials (#NCT02226627). All participants provided written informed consent. Individual confidentiality for the pooled data were maintained by using numeric identification codes rather than names. All data transfer was conducted with a secure file sharing system. 

### 2.2. Anthropometric Measurements

Anthropometric measurements were taken in duplicate by trained interviewers, according to the procedures proposed by the WHO [22]. The WHO reference was used to determine weight status in adults [22]: underweight (BMI < 18.5 kg/m^2^), normal weight (BMI = 18.5–24.99 kg/m^2^), overweight (BMI = 25.0–29.99 kg/m^2^), obesity (BMI ≥ 30–39.99 kg/m^2^), morbid obesity (BMI ≥ 40 kg/m^2^). Adolescents were classified according to the WHO z-scores for age and sex of each participant [23], considering <−2SD for thinness; >−2 to <+1SD for normal weight, >+1 to <+2SD for overweight, and >+2SD obesity.

### 2.3. Dietary Assessment

Dietary intake data were obtained from 2 face-to-face, 24 h recalls. Weekdays and weekend days were included with a proportional distribution of days among the sample to capture the day-to-day variations in food consumption. A photographic album of common foods of each country and household utensils were used to estimate portion sizes. Participants were asked to report all the foods and beverages consumed on the previous day, following the multiple-pass method [24]. Data were converted into energy, macro-, and micronutrients using the Nutrition Data System for Research software (NDS-R version 2013) [25]. Since NDS-R is based on the US Department of Agriculture food composition table, trained dietitians performed a previous standardization procedure to match local foods in each country in order to minimize errors [25]. Usual energy and nutrients intake were estimated for using the Multiple Source Method (MSM) (https://msm.dife.de/ (accessed on 16 July 2015)), which combines the probability and the amount of food consumed to estimate the usual intake for each individual [26].

Dietary quality score (DQS) was assessed following a methodology proposed by Imamura et al., 2015 [27]. Briefly, this approach evaluates the consumption of key dietary items adjusted for 2000 kcal per day, including healthy [i.e., fruits, vegetables, beans and legumes, nuts and seeds, whole grains, dairy, polyunsaturated fats (PUFAs), fish, plant omega 3, and dietary fiber] and unhealthy [i.e., unprocessed read meats, processed meats, sugar-sweetened beverages (SSB), saturated fat, trans fat, dietary cholesterol, and sodium] items. For each dietary factor mean, the intake was divided into age-, sex-, and country-specific quintiles. Each quintile was assigned an ordinal score. Higher scores were given to quintiles with higher mean intakes of healthy foods (from 1 to 5) for lower mean intakes of unhealthy foods. For unhealthy foods, higher scores were given to quintiles with lower mean intakes (from 5 to 1 points). All items were summarized and standardized to a 100-point scale in which the higher the scores, the healthier the diet. A detailed analysis of DQS in the ELANS study population was previously published [28]. All food items reported to have been consumed during the first 24 h recall were classified into 10 food groups to assess dietary diversity score (DDS), according to the Minimum Dietary Diversity Score for Women (MDD-W) proposed by FAO [29]: (1) starchy staples (grains, with roots and tubers and plantains); (2) meat, poultry, and fish; (3) dark-green leafy vegetables; (4) other vitamin A-rich fruits and vegetables; (5) other vegetables; (6) other fruits; (7) pulses (beans, peas, and lentils); (8) dairy; (9) eggs; and (10) nuts and seeds. For the consumption of at least 15 g/day, 1 point was assigned if consumed or 0 points if intake of that specific food group was less than 15 g/day, for a maximum score of 10 points. Micronutrient’s adequacy was determined using the Estimated Average Requirements (EAR) from the Institute of Medicine. They were used because they were a recommended standard parameter to estimate the prevalence of inadequate nutrient intake within a group [30]. Nutrient adequacy ratios (NAR) were estimated for vitamins A, C, D, and E, calcium, iron, thiamin, riboflavin, niacin, cobalamin, pyridoxine, zinc, magnesium, copper, phosphorus, and selenium. The NAR value for a given nutrient was the ratio of a participant’s current nutrient intake to the EAR for the corresponding sex and age category [31]. NAR’s values close to 1 implies that recommended nutrient intake was achieved or exceeded. Considering that a nutrient with a high NAR cannot be compensated by a nutrient with a low NAR, all NARs were truncated at 1. The mean adequacy ratio (MAR) was calculated as the sum of all NARs divided by the number of nutrients assessed.

### 2.4. Statistical Analyses

Data were analyzed using the Statistical Package for Social Sciences (SPSS) software program (version 23, SPSS Inc., Chicago, IL, USA). We classified the participants according to the level of alcohol intake using a cut-off criterium of ≥15 g/day of alcohol consumed calculated from the estimation of their usual daily intake. Thus, 2 subgroups were created: (1) alcohol consumers and (2) non-alcohol consumers. A chi-square test (χ^2^) was used to estimate the significant differences in the distribution of participants between countries, sex, and age intervals within the group of alcohol consumers.

The between-group comparisons of alcohol intake parameters (i.e., grams, calories, and percentage of calories from alcohol relative to the total energy intake) were analyzed by means of factorial multivariance analyses (MANOVA). For sociodemographic variables we used a two-way MANOVA followed by Bonferroni post-hoc test (when appropriate) with alcohol parameters as dependent variables and country, sex, socioeconomic status (SES), and age as fixed factors and subgroups of alcohol consumers as a second factor. One-way MANOVA was used to compare all dependent variables (e.g., macronutrients. food groups, micronutrients, diet quality indicators, and anthropometric measurements) between the 2 alcohol subgroups controlled by country, sex, and age. For the analysis of NAR scores, we also controlled by total energy intake as NAR values are strongly dependent of total energy consumed (see results for details).

To further provide evidence that the alcohol intake is related to nutritional and anthropometric variables, we classified the grams of alcohol into quartiles (Q) within the subgroup of alcohol consumers. Based on the grams of alcohol consumed within each quartile, we estimated the number of alcohol beverages representing each quartile as follows (Q0) 0 beverages, (Q1) 1 beverage, (Q2) 2–3 beverages, (Q3) 4–6 beverages, (Q4) more than 10 beverages. This classification allows us to identify the breaking points where alcohol consumption started to bestow health risks compared to non-consumers. Therefore, we reported the minimal number of beverages required to observe significant differences compared to non-consumers (e.g., 0 beverages) according to Bonferroni pairwise comparisons (all *p*-values < 0.05) after controlling by sex, country, and age. If there were significant differences between Q0 and Q1 or between Q0 and Q2 or Q3, the subsequent comparisons against Q0 were also significant.

We reported marginal means ± standard error of the mean (SEM) because these means were adjusted for the covariates included in the models and showed the exact values corresponding to the statistical analysis. The advantage of marginal means resides in the fact that they are estimated from complex models reflecting intricated effects that cannot be otherwise detected from the raw data. For that reason, the observed means and their standard deviation would not accurately represent the effects described by the models and, therefore, will slightly differ from the estimated marginal means. As an index of the effect size of the model, we reported the partial eta squared coefficients (η^2^_p_), which were also adjusted for the covariates included in the models. In all analyses, *p* < 0.05 values were considered statistically significant.

## 3. Results

### 3.1. Distribution of Sociodemographic Variables

There was substantial variability in alcohol intake in the whole sample (Table 1 and Appendix A). As 14–15 g of alcohol is commonly used to define a standard drink (e.g., a 12-ounces beverage with 5% of alcohol), we used a cut-off criterium of ≥15 g of alcohol to ensure that participants who met that criterium consumed at least the equivalent of one alcohol drink calculated as an estimation of their usual daily intake. Consequently, there were significant differences between the subgroups of consumers and non-consumers in the grams of alcohol consumed (*F*_(1,9202)_ = 6452.292, *p* = 0.00001, η^2^_p_ = 0.412), the net calories obtained from alcohol (*F*_(1,9202)_ = 6440.413, *p* = 0.0001, η^2^_p_ = 0.412), and the percentage of calories relative to the total energy intake (*F*_(1,9202)_ = 12,997.728, *p* = 0.0001, η^2^_p_ = 0.585). In the subgroup of alcohol consumers (Table 1), the mean alcohol intake was 69.22 ± 2.18 grams (range: 15–678 g), contributing with 484.62 kcal (range: 105–4793.46 kcal), which corresponds to 16.86% of the total energy intake (range: 2.36–69.64%). According to our cut-off criterium, there were 11.6% (*n* = 1073) of daily alcohol consumers in the whole ELANS sample. The non-consumers group (88.4%) comprised 6209 participants with 0 g of alcohol consumption, 1484 participants consuming between 0.005 and 1 g, and 452 participants consuming between >1 g and <15 g. Within the consumer’s groups, the consumption of alcohol beverages was distributed as follows: beer 803 persons, wine 194 persons, spirits 176 persons, and cocktails 53 persons. Although most participants drank more than one type of alcoholic beverages, there were more exclusive beer drinkers (83%) than those drinking only spirits (53%), cocktails (38%), and wine (27%).

The distribution of alcohol consumers and non-consumers differed significantly throughout the different sociodemographic variables, as shown in Table 1. There were differences in the distribution of consumers among countries (χ^2^_(7,9218)_ = 308.068, *p* = 0.0001), with Argentine (22.5%), Brazil (16.2%), and Chile (13%) having a percentage of consumer’s over 10% of the country sample. In contrast, Peru (7.5%), Venezuela (7.4%), and Ecuador (4.1%) had the lowest amount of alcohol consumers. Regarding the alcohol consumption, there was a main effect of Country (i.e., grams, calories, and percentage of calories: all *p*-values < 0.0001; η^2^_p_ = 0.064–0.098) and an interaction Country × Subgroups (i.e., grams, calories, and percentage of calories: all *p*-values < 0.0001; η^2^_p_ = 0.063–0.098). Specifically, in the subgroup of consumers, Ecuador showed the highest alcohol intake in grams and calories, which were higher than those of Argentine, Chile, and Costa Rica (Bonferroni, all *p*-values < 0.05). The second highest consumption (e.g., grams and calories) was detected in Venezuela, whereas the lowest consumption was observed in Argentine and Chile (Bonferroni, all *p*-values < 0.05). When comparing the percentages of alcohol calories, Venezuela and Brazil showed the highest proportion of calories derived from alcohol consumption, which was higher than that in Argentine, Chile, and Peru (Bonferroni, all *p*-values < 0.05). The lowest percentage of alcohol calories was observed in Argentine, which differed from all other countries except Chile and Peru (Bonferroni, all *p*-values < 0.05).

Regarding the other sociodemographic variables (Table 1), there were sex differences in the number of alcohol consumers, with men representing 68.3% of the consumer’s sample (χ^2^_(1,9218)_ = 204.175, *p* = 0.0001). In terms of alcohol intake, there was a main effect of Sex (i.e., grams, calories, and percentage of calories: all *p*-values < 0.0001; η^2^_p_ = 0.010) and an interaction Sex × Subgroups (i.e., grams, calories: all *p*-values < 0.0001; η^2^_p_ = 0.010) for grams and calories of alcohol, but not for its percentage. In the total sample, but especially in the subgroups of consumers, men consumed more alcohol than women. In addition, there were age differences (χ^2^_(3,9218)_ = 65.667, *p* = 0.0001) in the proportion of alcohol consumers in the whole sample, with the largest percentages of consumers (42.4%) being within 20–34 years, followed by those in the intervals of 30–49 years (31.3%) and 50–65 years (20.8%). Concerning the alcohol consumption, there was a main effect of Age (i.e., grams, calories, and percentage of calories: all *p*-values < 0.0001; η^2^_p_ = 0.004–0.010) and an interaction Age × Subgroups (i.e., grams and calories and percentage of calories: all *p*-values < 0.0001; η^2^_p_ = 0.004–0.010). In the whole sample, but especially in the alcohol consumers, the alcohol intake was almost the same in all age intervals excepting in the 50–65 age, in which both alcohol grams and calories were lower compared with all previous age intervals (Bonferroni, all *p*-values < 0.05). Regarding the percentage of alcohol calories, it was lower in the age of 15–19 years as compared with the following two age intervals (Bonferroni, all *p*-values < 0.05). Finally, the proportion of subjects consuming alcohol varied according to the socioeconomic status (SES) (χ^2^_(2,9218)_ = 774.257, *p* = 0.0001). The large percentage of consumers were in the low (48.6%) and middle SES (42.3%), with only 9.1% of consumers belonging to the high SES. There were also differences in the alcohol intake with a main effect of SES (i.e., grams, calories, and percentage of calories: all *p*-values < 0.0001; η^2^_p_ = 0.016–0.022) and an interaction SES x Subgroups (i.e., grams, calories, and percentage of calories: all *p*-values < 0.0001; η^2^_p_ = 0.009–0.014). In the subgroup of consumers, the participants in the low SES consumed more grams that corresponded to more net calories and a higher percentage of calories relative to the total energy intake as compared with the middle and high SES (Bonferroni, all *p*-values < 0.005), which did not differ to each other.

### 3.2. Anthropometric Parameters

As shown in Table 2, anthropometric parameters were compared between consumers and non-consumers after controlling by country, age, and sex as those factors have their own impact on body composition and weight. Consumers had higher body weight (*F*_(1,9212)_ = 10.614, *p* = 0.001, η^2^_p_ = 0.001), neck (*F*_(1, 9212)_ = 10.856, *p* = 0.001, η^2^_p_ = 0.001), hips (*F*_(1,9212)_ = 4.771, *p* = 0.029, η^2^_p_ = 0.001), and waist circumferences (*F*_(1,9212)_ = 34.548, *p* = 0.0001, η^2^_p_ = 0.004), with no differences on BMI, although it was descriptively higher in consumers. The analysis of the BMI categories revealed no significant differences in the proportion of subjects within each BMI category (data not shown). However, within the subgroup of consumers we found differences in the alcohol intake (grams and calories), especially when comparing the participants with normal weight versus those with morbid obesity (all *p*-values < 0.049; η^2^_p_ = 0.005) (Table 3).

### 3.3. Consumption of Kilocalories, Macronutrients, and Food Groups

The food consumption parameters are shown in Table 4. After controlling by country, age, and sex, we found that consumers exhibited a higher energy intake (*F*_(1,9212)_ = 1080.508, *p* = 0.0001, η^2^_p_ = 0.105), carbohydrates (*F*_(1,9212)_ = 30.055, *p* = 0.0001, η^2^_p_ = 0.003), fats (*F*_(1,9212)_ = 57.835, *p* = 0.0001, η^2^_p_ = 0.006), and proteins (*F*_(1,9212)_ = 78.317, *p* = 0.0001, η^2^_p_ = 0.008). When analyzing the percentage of calories derived from macronutrients, we found that consumers obtained less calories from carbohydrates (*F*_(1,9212)_ = 1747.413, *p* = 0.0001, η^2^_p_ = 0.159), proteins (*F*_(1,9212)_ = 701.275, *p* = 0.0001, η^2^_p_ = 0.071), and fats (*F*_(1,9212)_ = 550.686, *p* = 0.0001, η^2^_p_ = 0.056), but obtained on average 16.88% ± 0.35 calories from alcohol that non-consumers did not obtain (0.10% ± 0.001%. Regarding the food groups, we ranked them according to the effect size (i.e., percentage of variance explained by the alcohol intake) to highlight the eating pattern that differentiates the groups the most. Consumers ate a greater amount of some unhealthy food groups such as processed meat (*F*_(1,9213)_ = 62.248, *p* = 0.0001, η^2^_p_ = 0.007) and read meat (*F*_(1,9213)_ = 58.057, *p* = 0.0001, η^2^_p_ = 0.006). Although not significant, consumers also had a descriptively higher intake of SSB and added sugar. In contrast, consumers ate fewer of the healthy foods such as dairy products (*F*_(1,9213)_ = 42.620, *p* = 0.0001, η^2^_p_ = 0.005), fruits (*F*_(1,9213)_ = 32.308, *p* = 0.0001, η^2^_p_ = 0.003), fiber (*F*_(1,9213)_ = 26.494, *p* = 0.0001, η^2^_p_ = 0.003), whole grains (*F*_(1,9213)_ = 14.112, *p* = 0.001, η^2^_p_ = 0.002), and legumes (*F*_(1,9213)_ = 10.084, *p* = 0.0001, η^2^_p_ = 0.001). No significant differences were detected for fish, vegetables, and nuts.

### 3.4. Micronutrient Adequacy Ratio (NAR) and Diet Quality Indicators

The NAR values were compared between consumers and non-consumers after controlling by country, sex, age, and especially total energy intake because NAR values strongly depend on the amount of food consumed, and alcohol consumers had a higher caloric intake than non-consumers. In fact, all micronutrients’ intake correlated positively with energy intake irrespective of alcohol consumption, with Pearson’s correlation coefficients ranging from 0.038 to 0.567 (all *p*-values < 0.0001) (Table 5). In general terms, all NAR values were significantly lower in alcohol consumers (all *p*-values < 0.05). According to the effect size, the largest group differences were observed for vitamins E, A, and D, with eta square coefficients (η^2^_p_) ranging from 0.037 to 0.032, respectively. Other important differences were obtained for zinc, magnesium, vitamin C, and iron (η^2^_p_ = 0.019–0.012). Despite finding differences for all NAR values, the MAR score did not differ between groups, nor did the DQS. Significant differences were only observed for the DDS, which was lower in consumers (*F*_(1,9212)_ = 100.999, *p* = 0.0001, η^2^_p_ = 0.011).

### 3.5. Classification of Consumers According to Their Intake of Alcohol in Grams

We selected the grams of alcohol intake to classify in quartiles (Q) the participants of the subgroup of consumers as shown in Table 6. This classification allows us to identify the breaking points where alcohol consumption started to bestow risks of micronutrients inadequacy compared to non-consumers. We ranked the results to highlight the minimal number of beverages required to observe significant differences compared to non-consumers (e.g., 0 beverages).

For the anthropometric parameters (Table 7) to differ from non-consumers only the equivalent of 1 alcohol beverage was enough to increase body weight (*F*_(4,9198)_ = 11.857, *p* = 0.0001, η^2^_p_ = 0.005; Q0 vs. Q1–4). For the neck and waist circumferences more than 4 (*F*_(4,9198)_ = 29.120, *p* = 0.0001, η^2^_p_ = 0.013; Q0 vs. Q3–4) and 10 beverages (*F*_(4,9198)_ = 3.160, *p* = 0.013, η^2^_p_ = 0.001; Q0 vs. Q4) were rather needed to increase, respectively. As shown in (Table 6), the total energy intake (*F*_(4,9209)_ = 467.438, *p* = 0.0001, η^2^_p_ = 0.169; Q0 vs. Q1–4) increased with the equivalent of only 1 alcohol beverage as well as the reduction in the percentages of carbohydrates (*F*_(4,9209)_ = 548.750, *p* = 0.0001, η^2^_p_ = 0.192; Q0 vs. Q1–4) and proteins (*F*_(4,9209)_ = 259.539, *p* = 0.0001, η^2^_p_ = 0.101; Q0 vs. Q1–4). The percentage of energy from fats consumed reduced after 2 or more beverages (*F*_(4,9209)_ = 277.208, *p* = 0.0001, η^2^_p_ = 0.107; Q0 vs. Q2–4). The intake of proteins and carbohydrates increased after 2 or more beverages (*F*_(4,9209)_ = 23.457, *p* = 0.0001, η^2^_p_ = 0.010; Q0 vs. Q2-Q4) and 4 or more alcoholic beverages (*F*_(4,9209)_ = 18.077, *p* = 0.0001, η^2^_p_ = 0.008; Q0 vs. Q3-Q4), respectively. Regarding the food groups (Table 6), we found that the equivalent of only 1 alcoholic beverage was enough to increase the consumption of red meat (*F*_(4,9210)_ = 17.490, *p* = 0.0001, η^2^_p_ = 0.008; Q0 vs. Q1–4) and decrease the consumption of fiber (*F*_(4,9210)_ = 7.030, *p* = 0.0001, η^2^_p_ = 0.003; Q0 vs. Q1–4). For fruits (*F*_(4,9210)_ = 11.429, *p* = 0.0001, η^2^_p_ = 0.005; Q0 vs. Q2–4) and dairy products (*F*_(4,9210)_ = 15.708, *p* = 0.0001, η^2^_p_ = 0.007; Q0 vs. Q2–4) 2 or more alcoholic beverages already produced a reduction in the consumption, while for whole grains and vegetables the reduction appeared after 4 (*F*_(4,9210)_ = 4.999, *p* = 0.0001, η^2^_p_ = 0.002; Q0 vs. Q3–4) and 10 beverages (*F*_(4,9210)_ = 5.060, *p* = 0.0001, η^2^_p_ = 0.002; Q0 vs. Q4), respectively. No significant differences were observed in the consumption of fish and seafood and nuts and seeds.

The NAR values were inversely related to the alcohol consumption (Table 8), with significant differences being detected for all micronutrients analyzed. Magnesium (*F*_(4,9209)_ = 83.247, *p* = 0.0001, η^2^_p_ = 0.035; Q0 vs. Q1–4), and vitamin C (*F*_(4,9209)_ = 47.581, *p* = 0.0001, η^2^_p_ = 0.020; Q0 vs. Q1–4) decreased only after 1 beverage. Vitamin E (*F*_(4,9209)_ = 208.097, *p* = 0.0001, η^2^_p_ = 0.083; Q0 vs. Q2–4), vitamin A (*F*_(4,9209)_ = 161.864, *p* = 0.0001, η^2^_p_ = 0.066; Q0 vs. Q2–4), vitamin D (*F*_(4,9209)_ = 157.443, *p* = 0.0001, η^2^_p_ = 0.064; Q0 vs. Q2–4), zinc (*F*_(4,9209)_ = 93.626, *p* = 0.0001, η^2^_p_ = 0.039; Q0 vs. Q2–4), iron (*F*_(4,9209)_ = 51.266, *p* = 0.0001, η^2^_p_ = 0.022; Q0 vs. Q2–4), copper (*F*_(4,9209)_ = 40.965, *p* = 0.0001, η^2^_p_ = 0.017; Q0 vs. Q2–4), and thiamin (*F*_(4,9209)_ = 38.213, *p* = 0.0001, η^2^_p_ = 0.016; Q0 vs. Q2–4) showed a reduction after 2 or more beverages. For calcium (*F*_(4,9209)_ = 61.718, *p* = 0.0001, η^2^_p_ = 0.026; Q0 vs. Q2–4) and riboflavin (*F*_(4,9209)_ = 21.675, *p* = 0.0001, η^2^_p_ = 0.009; Q0 vs. Q2–4) 4 or mores beverages produced a reduction in the values, whereas for pyridoxin (*F*_(4,9209)_ = 24.141, *p* = 0.0001, η^2^_p_ = 0.010; Q0 vs. Q4), phosphorus (*F*_(4,9209)_ = 18.821, *p* = 0.0001, η^2^_p_ = 0.008; Q0 vs. Q4), cobalamin (*F*_(4,9209)_ = 10.545, *p* = 0.0001, η^2^_p_ = 0.005; Q0 vs. Q4), niacin (*F*_(4,9209)_ = 9.368, *p* = 0.0001, η^2^_p_ = 0.004; Q0 vs. Q4), and selenium (*F*_(4,9209)_ = 4.123, *p* = 0.002, η^2^_p_ = 0.002; Q0 vs. Q4) 10 or more beverages were sufficient to produce a reduction. Finally, the DDS (*F*_(4,9209)_ = 52.730, *p* = 0.0001, η^2^_p_ = 0.022; Q0 vs. Q3–4) reduced after 4 or more beverages.

## 4. Discussion

The main goals of the present study were first, describing the alcohol consumption of the ELANS sample, and second and foremost, estimating the role of alcohol consumption as an obesogenic factor affecting anthropometric parameters of obesity and nutritional status, which increase the risk of a myriad of health problems. To our knowledge, this is the first study evaluating alcohol consumption in relation to sociodemographic variables, dietary habits, and weight status among a large sample of the Latin American population.

In average, 11.6% of the population consumed at least one alcoholic beverage daily. Although the number of alcoholic drinks per day is not a sufficient criterium for diagnosing an alcohol use disorder, the consumption of more than 4 drinks per day (5.83%) constitute a pattern of heavy drinking that confer a higher risk for alcoholism and other psychiatric (e.g., neurocognitive, anxiety, and mood disorders) and non-psychiatric health problems (peripheric neuropathies, metabolic syndrome, obesity, and cardiovascular diseases). Those drinking 10 or more beverages per day are very likely to meet the criteria for alcohol use disorders (2.92%).

The pattern of alcohol consumption varied according to country, sex, age, and SES variables. In the whole sample, the alcohol drinkers consumed in average 4.6 beverages per day, although it ranged from 3.1 in Argentine and 7.7 in Ecuador, which showed the lowest and the highest number of alcohol drinkers, respectively. According to that number of alcoholic beverages per day, it is not surprising that Latin America and the Caribbean had the third-highest (6.7%) percentages of the population that met the criteria for an alcohol use disorder (AUD), surpassed only by North America (8.1%) and Europe (8.3%) [32]. Within Latin-American countries, Ecuador has the highest prevalence of AUD, which coincides with our data of Ecuador having the highest proportion of heavy drinkers.

On the other hand, we found a higher frequency of men drinkers compared to women (2.16:1 ratio), in agreement with previous studies consistently describing almost the same trend [6,15]. Although alcohol consumption has been progressively increasing in young women from middle and high SES [32], the gender gap in the proportion of drinkers is still present in Latin-American countries. This difference may be attributable to social constructs and gender roles, which have normalized alcohol consumption in men while have disapproved it in women as it has been considered as an immoral and unfeminine behavior [5,33]. In the subgroup of alcohol consumers, nevertheless, the differences between men and women in the number of alcoholic beverages consumed per day are rather small (5 versus 3.9 beverages), suggesting that the drinking patterns of the two genders are becoming more alike, especially in women consuming alcohol daily, who showed a moderate drinking habit. Interestingly, if alcohol consumption (e.g., grams of the number of beverages) is adjusted by body weight (e.g., kg), significant differences were no longer observed in these parameters, suggesting that women and men consumed almost the same amount of alcohol according to their body composition (data not shown). In the Americas, a small number of beverages consumed per day in men, and especially in women (4.3 versus 1.4), have been reported [32], which may be due to the use of a stricter cut-off criterion (e.g.,15 g/day) in our study that allowed us to describe only current and frequent consumers that have drunk at least one 12-ounce beverage with 5% of alcohol per day. Nor was there information in that study to estimate alcohol consumption relative to body weight, thus that the real dimensions of those gender differences in alcohol consumption are still unknown.

Regarding age, we observed the highest frequency of current alcohol drinkers in the range of 20–39 years, in agreement with previous reports [6,34,35,36]. However, in other world regions, such as in Nepal (5), higher odds ratios for alcohol consumption were observed in older adults (44–65 years old) belonging to disadvantaged ethnic groups, highlighting the geographic, cultural, and sociodemographic variations in the pattern of alcohol consumption worldwide. It is worth noting that there was 5.5% of adolescents or barely legal young adults (i.e., 15–19 years old) that consumed at least one alcoholic drink daily in our study. These data are not something to be taken lightly considering the age of the consumers. Furthermore, 51% of those young alcohol consumers were in the low SES, and only 10% were in the high SES, indicating that this population is at higher risk for developing an AUS due to its alcohol consumption pattern and its socioeconomic vulnerabilities. In fact, an early onset of alcohol use can shape the course of a problematic drinking pattern that substantially increases the risk for an AUS [37].

In regards to the latter, we found that there were more alcohol consumers in low SES irrespective of age, which also had the highest alcohol consumption in agreement with other studies [4,38,39]. The International Alcohol Study [40] demonstrated that drinkers below the poverty line across seven countries had a greater probability of being heavy drinkers, suggesting that the burden of heavier alcohol consumption is falling on people at the most vulnerable end of the socioeconomic spectrum. In contrast, other studies in Danish samples [4,41] found no significant role of income in determining drinking patterns. Data from more than 101,000 men and women from 33 countries of the GENACIS study found that higher SES was positively associateds with drinking status and that a higher country SES was associated with a higher proportion of drinkers [42]. However, it has been argued that even when people in the high SES consume similar or greater amounts of alcohol than those in the low SES, the less socioeconomically-disadvantaged individuals used to bear a disproportionate burden of negative alcohol-related consequences [39]. In addition, alcoholic beverages are expensive compared with traditional foods, which could lead to a reduction in healthy food purchases in those with limited income.

Previous ELANS findings have shown that poorer socioeconomic conditions were associated with less diet quality and diversity as well as fewer micronutrients adequacy ratios [43]. High alcohol intake is associated with less healthy dietary patterns [11]. Breslow et al. [44] found that the Healthy Eating Index-2005 scores significantly decreased with the consumption of alcoholic beverages based on the data from the US National Examination Survey (1999–2006). Even though we assessed diet quality using another approach, we have previously reported that diet quality in ELANS participants is poor in general [28]. In our current study, no significant differences in that score were observed between alcohol consumers and non-consumers. When assessing dietary diversity as a dimension of diet quality, we found that DDS decreased from 4.84 in non-alcohol consumers to 3.64 in alcohol-consumers in the fourth quartile (>10 beverages per day). Dietary diversity is widely recognized as a fundamental characteristic of a healthy diet [29], and monotonous diets have been related to micronutrients malnutrition in previous studies [31,45,46] and may be associated with the lower NAR scores observed among alcohol consumers in the present study.

In the current study, we found significant differences in food consumption and nutrient intake between alcohol drinkers and non-drinkers, which were proportionally higher according to the number of beverages consumed per day. In general terms, diet quality worsened as alcohol consumption increased, in agreement with previous research [47]. Energy intake was higher in alcohol consumers. In those consuming 10 or more beverages per day, the energy intake was 1.69-folds higher than that in non-consumers, supporting the association between the amount of alcohol consumed and the risk of surpassing the daily energy requirements that increase the likelihood of gaining weight. This also suggests that energy from alcoholic beverages is not well compensated by eating less food or substituting it for other foods, as mentioned elsewhere [11,48,49]. In heavy drinkers, alcohol consumption can represent up to 60% of total energy intake, which may lead to nutritional deficiencies and metabolic dysregulation, such as changes in the metabolism of carbohydrates and fiber, loss of protein mass, and the derangement of liver function [50]. Our data showed that alcohol drinkers consumed more carbohydrates, proteins, and fats than nondrinkers, but obtained significantly less energy from those macronutrients relative to the total energy intake. This proved to be even more true for heavy drinkers that obtained substantially less energy from macronutrients and more from alcohol. In contrast, there is evidence showing differences in the energy obtained from proteins and fats but no from carbohydrates in alcohol consumers [49], which highlights the variations in dietary patterns observed among countries.

Regarding food groups, we found that alcohol consumers, and especially moderate and heavy drinkers, consumed more red meat and less fiber, fruits, dairy products, whole grains, and vegetables. In line with our results, previous research has found lower intakes of fiber, fruits, and dairy products in alcohol drinkers and higher intakes of animal products as fish, poultry, meats, and animal fats [11,48]. The research developed in the Finish population found that moderate drinkers have higher energy intake from fats and proteins, lower consumption of fiber, dairy products, and fruits but higher intake of vitamin D, compared with non-drinkers [49,50].

Previous studies described that diet quality is deficient in alcohol drinkers [11,31,46]. We found that all NARs were significantly lower in alcohol consumers, with the NAR scores decreasing proportionally with the number of alcoholic drinks consumed per day. This effect of alcohol consumption on NAR values can be understood by two non-mutually exclusive explanations. First, alcoholic beverages provide a considerable number of calories but contain few or no micronutrients; and second, heavy alcohol consumption can lead to vitamins deficiency by reducing their absorption, altering their metabolism, or increasing their loss [51]. The largest shortfalls were observed for vitamins E, A, and D after consuming two or more alcoholic beverages per day. It is worth noting that the NAR scores for vitamin E were already low for the whole ELANS’ sample, but in alcohol consumers, the vitamin E levels were the lowest, meaning that in this population, oxidative stress from alcohol consumption and other sources may produce greater damage due to a weakened antioxidant system [52]. Besides energy and protein deficiency, chronic alcoholism has been associated with micronutrients inadequacy, including vitamin D, vitamin E, zinc, iron, and magnesium shortfalls, which can be caused by malabsorption due to cholestasis, pancreatic insufficiency, poor dietary intake, or metabolism impairments [52,53]. Regarding vitamin A, it has been reported that alcohol intake can alter retinoids homeostasis leading to a depletion of hepatic retinyl esters and retinol levels in chronic alcohol consumers, which can be linked to the development of alcoholic liver disease [54]. In addition, low serum vitamin D levels in alcohol users have been associated with cardiovascular diseases, hypertension, diabetes mellitus, metabolic syndrome, cancer, depression, schizophrenia, and other psychotic disorders [53]. Remarkably, for magnesium and vitamin C, only one alcoholic drink was enough to produce a reduction in the NAR levels compared to non-consumers. Two or more daily drinks already reduced the NAR scores for zinc, iron, copper, and thiamin, whereas differences for other micronutrients such as calcium and riboflavin could only be observed after 10 or more daily alcoholic beverages. Chronic alcohol consumption is known to decrease the absorption of thiamine and the activation of thiamine pyrophosphate, which acts as a coenzyme for many enzyme reactions in the metabolism of carbohydrates and amino acids and the subsequent production of energy. Impairment of that process in the nervous system is strongly associated with severe neurological conditions such as Wernicke-Korsakoff syndrome, alcoholic peripheral neuropathies, and the Marchiafava-Bignami disease [55]. Heavy drinkers have been found to have lower intake of several nutrients, including calcium and iron compared to non-drinkers [50] and may have less favorable micronutrients status compared with moderate and non-drinkers. Our findings should be interpreted with caution as this is a cross-sectional study that does not allow raising cause-and-effect inferences. Further studies exploring causal relationships between alcohol consumption and micronutrients shortfalls-induced health problems are warranted. Nevertheless, our results suggest that alcohol consumption might affect differentially the micronutrients status increasing the risk of an inadequate intake, which aggravates in heavy drinkers.

Alcohol has been suggested to stimulate further eating, besides adding calories to the total energy intake that are not compensated by eating less, non-alcoholic foods [56,57]. Alcohol-induced food consumption could be promoted by impairing inhibitory control leading to an overconsumption of energy-dense foods [12]. Additionally, alcohol can inhibit the action of satiety hormones such as leptin or glucagon-like petide-1, while increasing opioid, serotonergic, and GABAergic signaling pathways that stimulate appetite and potentiate the hedonic representation of foods, which reinforces cyclically the pattern of alcohol drinking, snacking, and binge eating [56]. Prior studies have shown a relationship between alcohol consumption and body mass index [48,58]. Our study demonstrated that body weight, waist, and neck circumference were significantly higher in alcohol consumers compared with non-consumers. Furthermore, alcohol consumers with normal weight consumed significantly less alcohol than those with morbid obesity. When analyzing the number of daily beverages needed to detect differences relative to non-consumers in the anthropometric parameters, we found that only one alcoholic beverage per day was enough to produce an increase in body weight, whereas 4 and 10 daily drinks were required to observe an increase in the circumference of neck and waist, respectively. In agreement with our results, others have also reported that heavier drinking is positively associated with BMI [6], waist circumference [6,10], and a higher risk for obesity [10]. Finally, one of the major strengths of our study is having a large, multinational representative sample of urban areas of eight Latin American countries following a standardized methodology that allowed a better comparison among them. Although rural areas were not represented, it is worth mentioning that the urban areas here included corresponded to more than 60% of the total Latin American population. Contrary to most studies of alcohol consumption, we also included a detailed analysis of nutritional and dietary patterns and anthropometric parameters of obesity. This study also had limitations as our results are based on two 24 h recalls of dietary intake. However, in order to minimize that bias, dietary data were collected by a trained interviewer using a standardized method, with the usual intake being estimated with the Multiple Source method that minimizes intra-individual variations in food consumption. Nevertheless, further studies may assess separately weekend and weekday recalls of dietary intake to better represent binge drinking patterns occurring at weekends.

## 5. Conclusions

The risk of alcohol consumption and its health consequences was higher in young and middle-aged men from low and middle SES. Alcohol consumption can be considered as an obesogenic factor as it increased anthropometric parameters of obesity (e.g., body weight, and neck, and waist circumference) and total energy intake, with macronutrients providing relatively less energy at expenses of the energy derived from alcohol. Alcohol drinkers characterized by having lower and higher consumption of healthy and unhealthy food groups, respectively. In addition, alcohol consumers showed an inadequate intake of micronutrients. All these deleterious effects of alcohol increased according to the number of alcoholic beverages consumed per day. Although it is well-known that alcohol dependence can increase the risk of dementia (e.g., Wernicke-Korsakoff type) and other neurocognitive disorders that are related to nutritional shortages, prevention, and treatment programs barely include nutritional components or evaluate the effects that alcohol and other drugs of abuse may have on nutritional status. To counteract those issues, we recommend capacitating public health personnel about the implications of heavy drinking on nutritional status and how that aggravates many health problems. In addition, it may be useful to implement educational programs in educative centers to provide simple but meaningful information at early ages about the risk of alcohol abuse and its consequences on the overall health, especially the effect of alcohol drinking on nutritional status. All this information should also be presented to decision-makers in government departments. As meeting micronutrients recommendations is essential to prevent chronic diseases, premature deaths, morbidity, and disability, additional efforts are required to ensure adequate vitamin and minerals intake in alcohol consumers. Thus, residential and medical centers aimed to treat drug dependence should consider including vitamins and minerals supplementation and personalized nutritional programs. Finally, moderating alcohol consumption is a prudent recommendation not only to reduce the risk of psychiatric, neurological, and cardiovascular diseases but also of obesity, metabolic syndrome, and diet-related diseases.

## Figures and Tables

**Table 1 ijerph-18-13130-t001:** Alcohol-consumption parameters according to the sociodemographic characteristics of alcohol consumers.

	*n*	Alcohol Consumption (g)/Day				Energy from Alcohol (Kcal)/Day				^1^ Energy from Alcohol (%)/Day			
		Mean ^2^	SEM	Min	Max	Mean	SEM	Min	Max	Mean	SEM	Min	Max
Overall sample	1073	69.22 (4.6)	2.17	15.00	678.00	484.62	15.27	105.00	4793.46	16.81	0.34	2.36	69.64
Sex													
Male	733	74.46 (5)	2.69	15.00	678.00	521.31	18.92	105.00	4793.46	17.00	0.42	2.36	69.64
Female	340	57.91 (3.9)	3.59	15.00	575.00	405.61	25.20	105.00	4065.25	16.54	0.61	3.73	68.88
Age group													
15–19 years	59	72.91 (4.9)	9.57	15.00	313.00	510.37	67.00	105.00	2191.00	15.40	1.49	2.36	56.83
20–34 years	455	73.55 (4.9)	3.35	15.00	564.00	514.88	23.50	105.00	3948.00	17.04	0.55	2.81	69.64
35–49 years	336	72.32 (4.8)	4.32	15.00	678.00	506.54	30.34	105.00	4793.46	17.61	0.62	3.50	67.89
50–65 years	223	54.71 (3.6)	3.65	15.00	470.00	383.01	25.57	105.00	3290.00	15.74	0.70	3.96	68.88
Countries													
Argentina	285	46.80 (3.1)	2.15	15.02	306.21	327.65	15.11	105.12	2143.49	11.64	0.41	2.81	49.64
Brazil	324	79.05 (5.3)	4.05	15.00	564.00	553.34	28.39	105.00	3948.00	21.05	0.69	2.36	68.88
Chile	115	50.61 (3.4)	4.58	15.00	306.00	354.29	32.06	105.00	2142.00	14.16	0.79	4.03	55.86
Colombia	81	92.10 (6.1)	11.51	15.00	575.00	645.24	80.87	105.00	4065.25	18.64	1.70	3.72	69.64
Costa Rica	67	64.64 (4.3)	6.80	15.00	346.00	452.49	47.63	105.00	2422.00	16.83	1.19	4.71	53.50
Ecuador	33	115.33 (7.7)	27.97	15.00	678.00	808.98	196.69	105.00	4793.46	17.99	23.6	3.73	55.26
Peru	84	67.44 (4.5)	5.78	16.00	257.00	472.09	40.51	112.00	1799.00	15.38	0.98	3.85	40.82
Venezuela	84	98.05 (6.5)	8.99	16.00	435.00	686.39	62.99	112.00	3045.00	21.42	1.30	3.96	58.35
Socioeconomic status													
Low	521	78.27 (5.2)	3.39	15.00	575.00	547.97	23.78	105.00	4065.25	18.58	0.54	2.81	68.88
Middle	454	62.93 (4.2)	3.17	15.00	678.00	440.64	22.28	105.00	4793.46	15.56	0.49	2.36	69.64
High	98	50.22 (3.3)	4.21	15.00	232.00	351.54	29.50	105.00	1624.00	13.71	0.79	4.56	38.60

Standard error of the mean (SEM). Grams of alcohol (g). Kilocalories derived from alcohol (Kcal). ^1^ Percentage of energy from alcohol [(alcohol energy/total energy intake) × 100]. ^2^ The numbers in parenthesis represent the equivalent of alcohol beverages per day according to the average consumption of grams of alcohol (grams of alcohol/15). See main text for details.

**Table 2 ijerph-18-13130-t002:** Anthropometric variables between consumers and non-consumers of alcohol.

Anthropometric Variables	Non-Consumers		Consumers	
	Mean	SEM	Mean	SEM
Neck circumference (cm)	35.56	0.40	35.95	0.11
Waist circumference (cm)	88.12	0.16	89.35	0.45
Hip circumference (cm)	100.29	0.13	101.12	0.36
Weight (Kg)	71.57	0.18	73.29	0.49
Body mass index (Kg/m^2^)	26.93	0.06	27.17	0.18

The mean and the standard error of the mean (SEM) were estimated based on multivariate variance analyses after controlling by country, age, and sex. See main text for details.

**Table 3 ijerph-18-13130-t003:** Alcohol intake among body mass index (BMI) levels within the group of alcohol consumers.

BMI Levels	Alcohol Consumption (g)/Day		Energy from Alcohol (Kcal)/Day		Energy from Alcohol (%)/Day	
	Mean	SEM	Mean	SEM	Mean	SEM
Normal weight	67.08	3.45	469.68	24.18	16.16	0.56
Overweight	67.20	3.42	470.42	23.95	16.83	0.56
Obese	74.64	4.87	522.67	34.15	18.05	0.79
Morbid obesity	98.72	15.61	691.00	109.43	19.97	2.53

Grams of alcohol (g). Kilocalories derived from alcohol (Kcal). The mean and the standard error of the mean (SEM) were estimated based on multivariate variance analyses after controlling by country, age, and sex. See main text for details.

**Table 4 ijerph-18-13130-t004:** Energy, macronutrients, and food groups according to alcohol consumption.

	Non-Consumers		Consumers	
	Mean	SEM	Mean	SEM
Energy (Kcal)	1980.96	6.39	2611.39	17.97
Carbohydrates (g)	270.58	0.04	286.02	2.64
Fats (g)	65.21	0.25	70.96	0.71
Protein (g)	78.04	0.24	84.41	0.68
Alcohol (g)	0.29	0.01	69.22	2.18
Carbohydrates (%)	54.42	0.08	44.63	0.22
Fats (%)	29.58	0.18	25.03	0.18
Protein (%)	16.01	0.03	13.40	0.09
Alcohol (%)	0.10	0.00	16.86	0.35
Wholegrains (g)	9.06	0.18	7.05	0.50
Fruits (g)	76.24	0.82	62.34	2.29
Vegetables (g)	105.48	0.59	106.93	1.65
Fish and seafood (g)	18.30	0.23	18.85	0.64
Nuts & seeds (g)	2.02	0.10	2.30	0.28
Beans & legumes (g)	37.81	0.42	33.86	1.17
Dairy products (g)	96.60	1.05	76.06	2.95
Sugar sweetened beverages (g)	677.59	5.12	683.86	14.39
Red meat (g)	63.45	0.38	72.21	1.07
Processed meat (g)	18.96	0.78	23.17	0.50
Added sugar (g)	65.30	0.40	67.13	1.10
Fiber (g)	15.87	0.06	14.94	0.17

Grams (g). Kilocalories (Kcal). The percentage from total energy intake (%). The mean and the standard error of the mean (SEM) were estimated based on multivariate variance analyses after controlling by country, age, and sex. See main text for details.

**Table 5 ijerph-18-13130-t005:** Nutrients Adequacy Ratio according to alcohol consumption.

	Non-Consumers		Consumers	
	Mean	SEM	Mean	SEM
Calcium	0.713	0.005	0.585	0.014
Iron	0.991	0.001	0.971	0.002
Vitamin D	0.368	0.002	0.254	0.006
Vitamin A	0.861	0.002	0.744	0.006
Vitamin C	0.865	0.002	0.779	0.007
Vitamin E	0.034	0.000	0.024	0.001
Thiamine	0.993	0.001	0.077	0.002
Riboflavin	0.990	0.001	0.981	0.002
Niacin	0.997	0.000	0.995	0.001
Pyridoxin	0.973	0.001	0.964	0.003
Cobalamin	0.968	0.001	0.978	0.002
Phosphorous	0.986	0.001	0.976	0.002
Magnesium	0.771	0.001	0.714	0.004
Zinc	0.970	0.001	0.931	0.003
Copper	0.990	0.001	0.971	0.002
Selenium	0.999	0.000	0.998	0.001
Mean Adequacy Ratio	0.814	0.001	0.816	0.002
Diet quality score	63.02	0.10	52.88	0.30
Dietary diversity	4.84	0.15	4.38	0.43

Micronutrient’s adequacy ratio (Mean consumption/Estimated Average Requirements (EAR)). The mean and the standard error of the mean (SEM) were estimated based on multivariate variance analyses after controlling by country, age, total energy intake, and sex. See main text for details.

**Table 6 ijerph-18-13130-t006:** Energy, macronutrients, and food groups consumption according to alcohol consumption quartiles.

	Consumers (*n* = 1073)									
	Q0 (*n* = 8145) 0 Beverage		Q1 (*n* = 268) 1 Beverage		Q2 (*n* = 268) 2–3 Beverages		Q3 (*n* = 268) 4–6 Beverages		Q4 (*n* = 269) >10 Beverages	
	Mean	SEM	Mean	SEM	Mean	SEM	Mean	SEM	Mean	SEM
Energy (Kcal)	1981.3	6.11	2195.7	33.83	2316.9	33.75	2566.4	33.83	3339.1	33.69
Carbohydrates (g)	270.59	0.93	267.51	5.18	272.63	5.17	294.11	5.18	309.20	5.16
Fats (g)	65.21	0.25	72.59	1.39	71.91	1.39	70.34	1.39	68.94	1.39
Proteins (g)	78.04	0.24	81.69	1.33	83.37	1.32	83.82	1.33	88.66	1.32
Alcohol (g)	0.29	0.01	20.64	0.21	34.89	0.32	60.01	0.70	160.94	5.50
Carbohydrates (%)	54.41	0.42	48.53	0.42	46.85	0.42	45.33	0.42	37.86	0.42
Fats (%)	29.58	0.06	29.41	0.35	27.51	0.35	24.35	0.35	18.91	0.35
Proteins (%)	16.00	0.03	15.00	0.18	14.50	0.18	13.17	0.18	10.93	0.18
Alcohol (%)	0.10	0.01	7.03	0.13	11.11	0.21	17.08	0.31	32.16	0.64
Wholegrains	9.07	0.18	8.58	0.99	7.69	0.99	6.51	099	542	0.99
Fruits	76.24	0.82	73.35	4.51	62.67	4.50	64.17	451	4937	4.50
Vegetables	105.48	0.50	116.62	3.26	108.99	3.26	105.63	326	96.63	3.25
Fish and seafood	18.29	0.23	17.58	1.27	18.33	1.26	17.87	126	21.53	1.25
Nuts and seeds	2.02	0.10	1.94	0.55	2.43	0.55	2.95	055	1.88	0.55
Beans and legumes	37.81	0.42	29.40	2.30	29.57	2.29	34.68	229	41.68	2.29
Dairy products	96.59	1.05	93.55	5.81	82.43	5.79	68.98	580	59.52	5.78
Sugar sweetened beverages	677.58	5.12	672.05	28.34	688.28	28.28	760.48	280.30	615.47	28.24
Red meat	63.46	0.38	71.02	2.11	69.09	2.12	70.31	2.12	78.36	2.11
Processed meats	18.97	0.18	22.96	0.98	23.02	0.98	20.79	0.98	25.87	0.98
Added sugar	65.30	0.40	66.46	2.17	67.00	2.17	72.78	2.17	62.33	2.17
Fiber	15.87	0.06	14.98	0.33	14.88	0.33	15.24	0.33	14.65	0.33

Grams (g). Kilocalories (Kcal). Percentage from total energy intake (%). Grams of alcohol were split into quartiles (Q) and the mean values of each quartile was divided by 15 to obtain the equivalent number of beverages per day. The mean and the standard error of the mean (SEM) were estimated based on multivariate variance analyses after controlling by country, age, and sex. See main text for details.

**Table 7 ijerph-18-13130-t007:** Anthropometric measurements according to alcohol consumption quartiles.

	Alcoholic Consumers (*n* = 1073)									
	Q0 (*n* = 8145) 0 Beverage		Q1 (*n* = 268) 1 Beverage		Q2 (*n* = 268) 2–3 Beverages		Q3 (*n* = 268) 4–6 Beverages		Q4 (*n* = 269) >10 Beverages	
	Mean	SEM	Mean	SEM	Mean	SEM	Mean	SEM	Mean	SEM
Neck circumference (cm)	35.47	0.04	36.28	0.24	36.11	0.24	36.92	0.23	37.58	0.24
Waist circumference (cm)	88.13	0.17	88.27	0.81	88.71	0.81	89.38	0.81	90.75	0.80
Hip circumference (cm)	100.42	0.13	99.65	0.69	100.67	0.69	100.10	0.69	100.08	0.69
Weight (kg)	71.37	0.18	73.37	0.98	74.60	0.98	74.68	0.98	76.48	0.98
Body Mass Index (Kg/m^2^)	26.99	0.06	26.42	0.33	26.60	0.33	26.87	0.33	27.15	0.33

Grams of alcohol were split into quartiles (Q) and the mean values of each quartile was divided by 15 to obtain the equivalent number of beverages per day. The mean and the standard error of the mean (SEM) were estimated based on multivariate variance analyses after controlling by country, age, and sex. See main text for details.

**Table 8 ijerph-18-13130-t008:** Nutrients Adequacy Ratio and Dietary Diversity Score according to alcohol consumption quartiles.

	Alcohol Consumers (*n* = 1073)									
	Q0 (*n* = 8145) 0 Beverage		Q1 (*n* = 268) 1 Beverage		Q2 (*n* = 268) 2–3 Beverages		Q3 (*n* = 268) 4–6 Beverages		Q4 (*n* = 269) >10 Beverages	
	Mean	SE	Mean	SE	Mean	SE	Mean	SE	Mean	SE
Calcium	0.715	0.005	0.755	0.026	0.684	0.026	0.549	0.026	0.296	0.028
Iron	0.991	0.001	0.990	0.003	0.98	0.003	0.963	0.003	0.947	0.003
Vitamin D	0.369	0.002	0.356	0.001	0.298	0.011	0.241	0.011	0.086	0.012
Vitamin A	0.862	0.002	0.837	0.011	0.798	0.011	0.74	0.011	0.567	0.012
Vitamin C	0.865	0.002	0.828	0.013	0.794	0.013	0.771	0.013	0.705	0.014
Vitamin E	0.034	0.000	0.032	0.001	0.03	0.001	0.023	0.001	0.006	0.001
Thiamine	0.993	0.001	0.991	0.003	0.982	0.003	0.971	0.003	0.960	0.003
Riboflavin	0.990	0.001	0.993	0.003	0.987	0.003	0.981	0.003	0.959	0.003
Niacin	0.997	0.000	0.999	0.002	0.998	0.002	0.995	0.002	0.985	0.002
Pyridoxin	0.973	0.005	0.985	0.005	0.975	0.005	0.964	0.005	0.924	0.005
Cobalamin	0.986	0.001	0.990	0.005	0.984	0.005	0.980	0.005	0.054	0.005
Phosphorous	0.986	0.001	0.987	0.004	0.987	0.004	0.977	0.004	0.95	0.004
Magnesium	0.772	0.007	0.743	0.007	0.741	0.007	0.721	0.008	0.631	0.008
Zinc	0.970	0.001	0.968	0.005	0.946	0.005	0.924	0.005	0.874	0.005
Copper	0.990	0.001	0.987	0.003	0.978	0.003	0.967	0.003	0.948	0.004
Selenium	0.999	0.000	1.000	0.001	0.999	0.001	0.998	0.001	0.994	0.001
Dietary Diversity Score	4.847	0.014	4.725	0.079	4.63	0.079	2.373	0.080	3.647	0.085

Micronutrient’s adequacy ratio (Mean consumption/Estimated Average Requirements (EAR)). Grams of alcohol were split into quartiles (Q) and the mean values of each quartile was divided by 15 to obtain the equivalent number of beverages per day. The mean and the standard error of the mean (SEM) were estimated based on multivariate variance analyses after controlling by country, age, total energy intake, and sex. See main text for details.

## Data Availability

Due to ethical and legal restrictions of the eight institutions involved, the data underlying this study are available upon request and must be approved by the Publishing Committee of ELANS. To apply for access to these data, interested researchers must submit a detailed project proposal. Requests for the data can be made to the correspondence author.

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
