# Peer review of "Alcohol Contribution to Total Energy Intake and Its Association with Nutritional Status and Diet Quality in Eight Latina American Countries"

_ijerph, 2021, doi:10.3390/ijerph182413130_

Round 1

Author Response

REVIEWER 1

First comment

Thank you for your comments. We clarified that currently in population studies involving large numbers of individuals it is proposed to use short-term dietary methods such as 24-hour recall. It has been recommended to use at least two 24-hour measures in 40% of the population to minimize errors and better estimate the habitual dietary pattern. However, our study used two, 24-hour recalls in the whole sample. We also used the Multiple Source Method (MSM) for adjusting for intrapersonal variability and obtaining usual intake (Harttig, 2011).

The methodological choice validates the use of the two 24-hour recalls applied on non-consecutive days which representing every day of the week and all seasons of the year. This represents the usual intake of alcohol in the study population. When applying the MSM method, co-variables such as sex, age, and an indicator of first-day versus second-day dietary recall are included to account for sequence effects of a subject's dietary recall.

To clarify the reader of the article, information highlighted earlier in the text of the article has been inserted:

Line 162- Data were collected on non-consecutive days, representing every day of the week.

Lines 172- The usual intake was estimated using the Multiple Source method, which combines the probability and the amount of food consumed to estimate the usual intake for each individual (1).

The limitations regarding the instrument and how they were minimized are included in the discussion of the paper.

One of the major strengths of our study is having a large, multinational representative sample of urban areas of eight Latin American countries following a standardized methodology that allowed a better comparison among them. Contrary to most studies of alcohol consumption, we also included a detailed analysis of nutritional and dietary patterns and anthropometric parameters of obesity. This study also had limitations as our results are based on two 24-h recalls of dietary intake. However, in order to minimize bias, dietary data were collected by trained interviewers using a standardized method, usual intake was estimated by using two 24HR and the Multiple Source method.

Abstract

Corrected.

Introduction

All observations were incorporated. 

Methods

All observations were incorporated. 

Results

As requested, we added text before the table (1) and also the proportion of participants with zero intake (2), but we did it in text in the main manuscript instead of including a supplementary figure (e.g., histogram) because it may be useful in the section in which we included it.

(3) The table 1 shows only the consumers (n=1073). There is a supplementary table with the non-consumers (n=8145). This may explain the differences in the sample size reported there.

(4) We did not reduce the decimals because last time the journal asked us to add them to all data presented consistently.

(5) We clarified all the details in tables that were not sufficiently well explained, including other aspects not mentioned by the Reviewer. The number in parenthesis are the estimated number of beverages per day according to the mean grams for each category. Now it is well explained. We refused to add p-values because the tables were already overloaded of details and in the main text we were already reporting not only the p-values, but also the size effect.

(6) We are aware about the number of tables, but we tried different options and ended up keeping the tables as they provided more details and looked cleaner without consuming too much space in the manuscript. Except for table 2 and 3, the rest of the tables have 19 or more rows and 3 or more columns, which when converted to graphs (bars, pies or circos charts), turned out to be less practical to the reader.

Discussion and conclusion

(1) Regarding the discussion, we did not remove percentages or odd ratios because we felt they provide context to discuss the data, especially when compared to other studies. In most cases, the numbers given in the discussion were not mentioned in the results section in the same way.

(2) The suggestions about the moving paragraphs or elaborating a bit more about certain topics were all addressed.

(3) Spelling mistakes were all corrected, excepting the suggested change of male for men. As far as we understand, the right gender pair is women and men or female and male, but not male and women. By doing a quick surfing, we did not come across with that combination.

Reviewer 2 Report

There is a well done wrote manuscript.

A very well-described manuscript with a very large number of respondents (N=9,218!) and, above all, multi-center cooperation, which confirms the fact of observing the consumption of all products, including alcohol.

I suggest the next time observation of 3 or 7 days and on maybe a smaller number, which will also be able to show interesting eating behavior.

References are up to date and properly selected. However, I suggest a few corrections:

Introduction:

  1. Please add information on how much kcal is in 1 g of pure ethanol.

Materials and methods:

  1. I would like to know what days you watched/analyzed in your questionnaire?
  2. Why did the authors decide to analyze only 24h?
  3. Why did the authors decide to analyze only this part of Latin America? Maybe next time it could be extended to other countries. Because it is worth continuing this direction of research.
  4. Could you also add information on what kind of alcohol the participants were drinking?
  5. Do you have any information on the listed socioeconomic status and any differences without these groups?

Results:

  1. I suggest presenting some data in the diagram which will be more reliable for the readers.

Discussion:

  1. The reader may be interested in the loss of vitamins and/or trace elements after long-term alcohol consumption.

Conclusion:

  1. I would suggest adding a few recommendations for public institutions, all medical personnel who work with addicts. Maybe also teachers, students, and government offices/departments.
  2. Maybe it is also worth changing some education in high school or even elementary school. To sum up, after explaining the strengths and weaknesses in the text, I recommend publishing the article after the above changes.

Author Response

REVIEWER 2

Introduction

Information about ethanol kcal was included.

Materials and methods

I would like to know what days you watched/analyzed in your questionnaire?

This information was added.

Why did the authors decide to analyze only 24h?

We analyzed 24-h recall to reduce operational costs in such a large simple, because there were face-to-face interviews. To compensate, we did two recalls on non-consecutive days and used the Multiple Source Method (MSM) adjusted for intrapersonal variability to obtaining usual intake for each individual.

Why did the authors decide to analyze only this part of Latin America? Maybe next time it could be extended to other countries. Because it is worth continuing this direction of research.

The countries included in ELANS where selected because all countries accomplished the requirements to follow a common study protocol for interviewers training, implementations of fieldwork, data collection, and quality control procedures. Each country had an academic-based principal investigator that facilitated the above mention procedures. At the time data were collected, some others Latin American countries have recently performed National Nutrition Surveys, so they considered this study as not necessary at that moment. The sample included in this study represents approximately 60% of the total Latin America population.

Could you also add information on what kind of alcohol the participants were drinking?

We added that information.

Do you have any information on the listed socioeconomic status and any differences without these groups?

We do not have more information regarding the socioeconomic status other than the one presented in the study.

Results

I suggest presenting some data in the diagram which will be more reliable for the readers.

We are aware about the number of tables, but we tried different options and ended up keeping the tables as they provided more details and looked cleaner without consuming too much space in the manuscript. Except for table 2 and 3, the rest of the tables have 19 or more rows and 3 or more columns, which when converted to graphs (bars, pies or circos charts), turned out to be less practical to the reader.

Discussion and conclusion

We elaborated on the effects of long-term alcohol consumption on micronutrients and the likely consequences of their shortfalls on health.

Both in the discussion and especially in the conclusions, we also added some few recommendations at the different levels suggested by the Reviewer.
